# Rapid Extraction of Regional-scale Agricultural Disasters by the Standardized Monitoring Model Based on Google Earth Engine

**Zhengrong Liu [1], Huanjun Liu [1,2], Chong Luo [2], Haoxuan Yang [3] , Xiangtian Meng [1], Yongchol Ju [4] and Dong Guo [2,\*]**

[1]   School of Pubilc Adminstration and Law, Northeast Agricultural University, Harbin 150030, China; liuzhengronghxs@gmail.com (Z.L.); liuhuanjun@neigae.ac.cn (H.L.); liuzhao6699@gmail.com (X.M.)
[2]   Northeast Institute of Geography and Agroecology Chinese Academy of Sciences, Changchun 130102, China; luochong@iga.ac.cn
[3]   College of Surveying and Geo-Informatics, Tongji University, 1239 Siping Road, Shanghai 200092, China; yhx965665334@gmail.com
[4]   Wonsan University of Agriculture, Won San City, Kangwon Province, DPRK; yuezou929@gmail.com
\*   Correspondence: guodong@iga.ac.cn; Tel.: +86-1874-519-4393

**Abstract:** Remote sensing has been used as an important tool for disaster monitoring and disaster scope extraction, especially for the analysis of spatial and temporal disaster patterns of large-scale and long-duration series. Google Earth Engine provides the possibility of quickly extracting the disaster range over a large area. Based on the Google Earth Engine cloud platform, this study used MODIS vegetation index products with 250-m spatial resolution synthesized over 16 days from the period 2005–2019 to develop a rapid and effective method for monitoring disasters across a wide spatiotemporal range. Three types of disaster monitoring and scope extraction models are proposed: the normalized difference vegetation index (NDVI) median time standardization model ($R_{NDVI\_TM(i)}$), the NDVI median phenology standardization model ($R_{NDVI\_AM(i)(j)}$), and the NDVI median spatiotemporal standardization model ($R_{NDVI\_ZM(i)(j)}$). The optimal disaster extraction threshold for each model in different time phases was determined using Otsu's method, and the extraction results were verified by medium-resolution images and ground-measured data of the same or quasi-same period. Finally, the disaster scope of cultivated land in Heilongjiang Province from 2010–2019 was extracted, and the spatial and temporal patterns of the disasters were analyzed based on meteorological data. This analysis revealed that the three aforementioned models exhibited high disaster monitoring and range extraction capabilities, with verification accuracies of 97.46%, 96.90%, and 96.67% for $R_{NDVI\_TM(i)}$, $R_{NDVI\_AM(i)}$, and $_{(j)}R_{NDVI\_ZM(i)(j)}$, respectively. The spatial and temporal disaster distributions were found to be consistent with the disasters of the insured plots and the meteorological data across the entire province. Moreover, different monitoring and extraction methods were used for different disasters, among which wind hazard and insect disasters often required a delay of 16 days prior to observation. Each model also displayed various sensitivities and was applicable to different disasters. Compared with other techniques, the proposed method is fast and easy to implement. This new approach can be applied to numerous types of disaster monitoring as well as large-scale agricultural disaster monitoring and can easily be applied to other research areas. This study presents a novel method for large-scale agricultural disaster monitoring.

**Keywords:** Google Earth Engine; MODIS; disaster monitoring; remote sensing index; NDVI standardization model

## 1. Introduction

Climate impact and environmental change are two important factors that restrict the development of agricultural production. Among them, the impacts of droughts, windstorms, pest infestations, hailstorms, and other agricultural disasters are the most significant. As a result of the global warming trend, the increasing frequency and intensity of various extreme weather events around the world have brought great harm to food security and agricultural development [1]. The traditional agricultural disaster monitoring methods are time-consuming and mainly consist of field investigation and sampling, which are difficult to implement in large areas. Compared with the traditional methods, the use of remote sensing to monitor agricultural disasters has the advantages of continuous spatiotemporal access to high-resolution surface information, fast data acquisition, and a wide range. For these reasons, remote sensing has been widely used in agricultural disaster and vegetation dynamic monitoring, and numerous remote sensing measurement methods have been developed to monitor global vegetation and extreme climate events [2,3]. The monitoring of agricultural disasters via remote sensing plays an essential role in rapid crop loss assessment, crop condition monitoring, crop insurance, and food security. Therefore, there is an urgent necessity to establish a rapid and large-scale agricultural disaster monitoring method with remote sensing as its technical basis.

At present, many agricultural disaster monitoring methods have been proposed, including ground spectral features, remote sensing vegetation indices, and vegetation index time series. Many vegetation indices based on remote sensing parameters, including the normalized difference vegetation index (NDVI), enhanced vegetation index (EVI), normalized difference water index (NDWI), vegetation condition index (VCI), vegetation health index (VHI), disaster vegetation damage index (DVDI), fire weather index (FWI), crop water stress index (CWSI), vegetation supply water index (VSWI), and temperature vegetation dryness index (TVDI), are widely used in disaster monitoring. Furthermore, based on these indices, a daily-scale forest fire danger forecasting system (FFDFS) was developed for drought monitoring, and a fire risk assessment and remote sensing-based flood crop loss assessment service system (RF-CLASS) has been employed to assess crop damage caused by waterlogging [4–10]. The VCI has proven to be an effective means of monitoring drought occurrence and measuring the intensity, duration, and impact of droughts around the world. The spatial and temporal ranges of agricultural drought can be studied via the VCI [11], although the correlation between the VCI and the meteorological drought index based on weather station data is not high [12]. The VCI is also not very sensitive to short-term precipitation shortages. In addition, there is significant spatial variability in the relationship strength between the VCI and the meteorological drought index [13]. The VHI is a widely used comprehensive remote sensing drought index whose goal is to improve the VCI in areas with high soil moisture and long-term cloudy conditions [14]. It is also used to evaluate the degree of agricultural drought and extract the spatiotemporal range of drought [15]. However, drought monitoring via the VHI requires the assumption of a negative correlation between the NDVI and land surface temperature (LST). Therefore, the VHI is not applicable in regions and periods where the NDVI-Ts (surface temperature) correlation coefficient is non-negative [16]. The TVDI is feasible for large-scale drought monitoring, although it is usually affected by its high sensitivity to clouds. Hence, it should not be used to monitor moderate and severe droughts [17–19]. The crop water stress index (CWSI) is widely used as an indicator of crop water status. The short-term oscillations of canopy temperature and vegetative flushing are the main factors that make the CWSI less effective in wet areas, which is its chief limitation [20,21]. The VSWI and TVDI can be used for drought monitoring, but they are not suitable for areas with large elevation changes [22]. Moreover, the CWSI, TVDI, and VSWI exhibit certain lags in drought detection, meaning that they take some time to respond [23]. In view of these lagging vegetation indices, hyperspectral remote sensing technology can be used to monitor winter wheat freezing injury and locust disasters [24–26]. The DVDI, which is often used in flood disaster and wind disaster monitoring, has a linear relationship with crop yield reduction and is an effective indicator of the degree of vegetation damage [27,28]. At the same time, the EVI is also frequently utilized to describe vegetation patterns in ecosystems affected by hurricanes,

such as tropical rainforests, tropical arid forests, and temperate arid grasslands. The NDVI and EVI, as the most widely used remote sensing indicators, are usually adopted for crop growth monitoring. MODIS NDVI time series can be used to analyze the spatiotemporal evolution of droughts and ENSO events in order to estimate the associated yield loss [29–31]. In areas with less vegetation, methods based on the vegetation index have their limitations. For desert locusts, based on the mid-infrared (MIR), near-infrared (NIR), and red reflectance, multi-temporal and multi-spectral image analyses are effective [32]. Corn fields damaged by hail can be effectively identified by comparing the ΔNDVI before and after the hail from HJ-1 CCD images, although it is difficult to precisely classify the damage [33,34]. Pixel-based time series derived from enhanced vegetation index (EVI) data can be extracted to detect flood disturbances of crop production, but when assessing flood events occurring during crop maturity, the accuracy rate is very low [35]. At the same time, the habitat of Asian locusts can be monitored [36]. Some studies have employed three different remote sensing green indices, namely the normalized vegetation difference index (NDVI), the enhanced vegetation index (EVI), and the green index (GI), to study the damage of frost to the canopy [37]. The above indicators have been widely used to monitor crop growth in specific regions and countries, as well as the entire world. Crop growth monitoring usually uses the NDVI as the main indicator of crop conditions, either by combining the NDVI value with other variables for analysis and utilization, or by calculating the difference between the multi-year average (or selected "reference" year) and the NDVI of that year to monitor the growth of crops [38]. However, this method also has its limitations. First, one needs to obtain multi-year averages for the same crop, which requires that the crop planting structure and distribution remain unchanged. Second, the error of the crop growth fluctuation in the selected reference year will affect the assessment results of that year. In order to avoid the crop distribution changes that lead to information errors, C. Li proposed monitoring the growth of winter wheat based on the percentage of crop NDVI (pNDVI) [39]. Few studies, however, can remove this limitation in terms of phenology. It is also very important to use NDVI data based on geographic information system (GIS) environment tools to monitor agricultural disasters. Remote sensing (RS) and GIS have been integrated to monitor and forecast rice production in Bangladesh [40], and have also been utilized to study the influence of mining on the surface temperature and vegetation conditions [41].

In addition, most research generally focuses on relatively small areas. For example, when monitoring disasters using ground spectral characteristics, the use of visible and near-infrared reflectance spectroscopy is an alternative method for monitoring soil contaminated by heavy metals, although the study area tends to focus on either a particular city or county [42,43]. From the above research, we determined that the traditional disaster monitoring methods rely on the disaster data collected by surface stations in order to construct indicators based on the data. In addition, the amount of data obtained is limited, and the data are difficult to collect. Moreover, many disaster monitoring methods based on remote sensing exhibit various application shortcomings. Most disaster monitoring research methods are limited by large image data, generally focus on small time scales or small research areas, and their speeds are slow, since they lack a fast, large-scale disaster discrimination scheme. Therefore, it is difficult to quickly analyze the spatiotemporal disaster pattern in a certain area. Google Earth Engine (GEE) can solve this problem, since it can quickly carry out large-scale and long-range disaster monitoring in a long-term sequence to analyze the spatiotemporal pattern of the designated area. GEE is a cloud platform that stores and processes BP-level global time series satellite images and vector data. Researchers from various countries have used GEE to conduct research on vegetation monitoring, land cover, agricultural applications, disaster management, and Earth science [44,45]. In cropland mapping, most of the previous studies were limited to the agricultural production within a given period, growing season, or year. Cloud computing based on GEE, however, can quickly and automatically cover a large range of farmland [46,47]. At the same time, its powerful computing ability does not limit research to particular agricultural fields, thus providing the possibility of depicting the rice planting area map for Northeast Asia and the fallow paddy area for the Sanjiang Plain [48,49]. GEE also plays an important role in yield mapping and crop classification [50,51].

Meanwhile, a gradually increasing number of studies on disasters have been carried out by researchers using GEE, such as A. Beaton et al., who calculated the ice breakup period of a river for flood monitoring using GEE [52]; N. Sazib et al., who verified the value of global soil moisture data for drought disaster monitoring using GEE [53]; C. C. Liu et al., who developed a flood control and emergency system based on GEE (FPERS) [54]; and B. Pradhan et al., who used GEE to provide physical support for the assessment of the forest impacts of sand dune risk and hurricanes in the Sabha region of Libya [55]. Based on GEE, L. Lu et al. examined the spatial characteristics of vegetation destruction induced by typhoons in the coastal areas of southeastern China from 2000 to 2018 [56]. However, since most of these disaster assessments focused on single disasters, there remains a lack of large-scale disaster monitoring and range extraction methods for crops affected by multiple disasters.

In terms of phenology, there have been few studies on large-scale disaster monitoring and disaster range extraction. In addition, most of the research has focused on a single disaster type and has lacked a method for extracting a wide range of disaster types. In GEE, different vegetation indices extracted through multi-temporal remote sensing images are used as standard values to reflect the normal conditions of crop growth in different regions and different growth stages, and they are compared with the vegetation index extracted in a single time phase in order to compare agricultural disasters in the region. The situation is monitored more accurately, thereby making the results universally applicable. It remains difficult, however, to extract the standard value that can represent the average growth of crops. To address this issue, this study introduces the NDVI median time normalization model ($R_{NDVI\_TM(i)}$), the NDVI index median phenology standardization model ($R_{NDVI\_AM(i)(j)}$), and the NDVI median time-space normalization model ($R_{NDVI\_ZM(i)(j)}$), which comprehensively consider the effects of phenology, different disasters and crop types, and changes in the planting structure, with the goal of proposing a large-scale GEE-based monitoring method for the rapid extraction of agricultural disasters. We attempted to utilize the MODIS 16-day NDVI time series data after smooth reconstruction and compare and analyze the regional-scale disaster index analysis map generated by the three models. Additionally, we planned to extract the disaster threshold of the study area using Otsu's method and compare it with HJ-1A/B CCD data in order to analyze the spatial and temporal distributions of disasters in Heilongjiang Province from 2000 to 2019. This method features good transferability and can be quickly applied in other areas.

## 2. Materials and Methods

### 2.1. Study Area

Located between latitude 43°25′–53°33′ N and longitude 121°11′–135°05′ E, Heilongjiang Province straddles three humidity zones from east to west. The total land area of the province is approximately 473,000 km$^2$, of which agricultural land accounts for ~39.5045 million hectares. In terms of elevation, it is high in the northwest, north, and southeast, and low in the northeast and southwest (Figure 1). Heilongjiang Province is located in the eastern part of Eurasia, to the west of the Pacific Ocean, and has a temperate continental climate. The average annual temperature in the province generally ranges from −5 to 5 °C, and its annual precipitation varies from 400 to 650 mm, with uneven spatial and temporal distributions. Agricultural disasters are frequent, and the frequencies of the representative disasters of droughts, floods, windstorms, hailstorms, low temperatures, and freezing, as well as disease and insect disasters, are increasing [57,58].

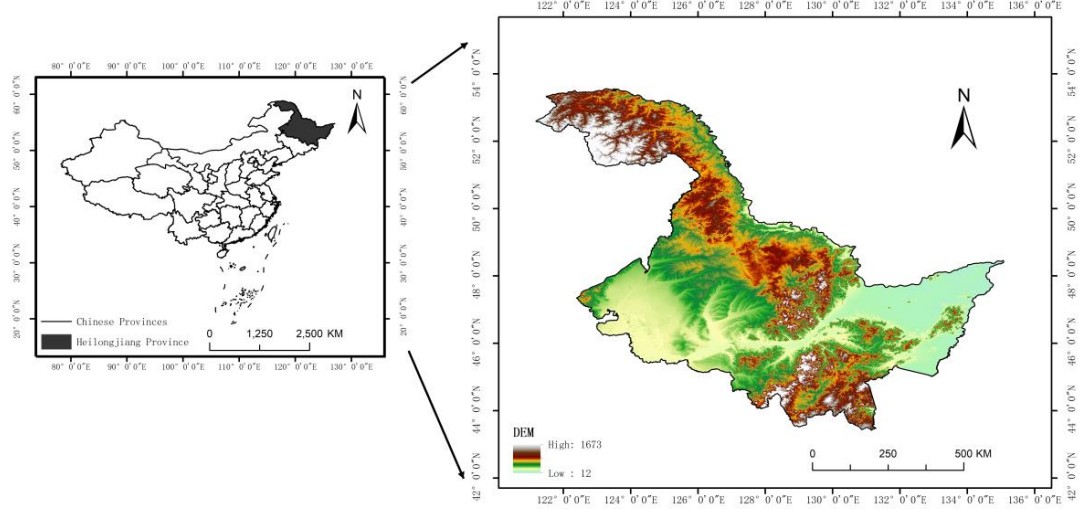

**Figure 1.** Elevation map of Heilongjiang Province.

*2.2. Data*

2.2.1. MOD13Q1

The MODIS vegetation index (MOD13Q1) synthesized over 16 days with a 250-m spatial resolution that was used in the study area is a terrestrial data product, whose complete and formal designation is MODIS/Terra Vegetation Indices 16-day L3 Global 250-m SIN Grid (Table 1). This product is calculated by the atmospheric correction of bidirectional surface reflectance and possesses the advantages of moderate spatial resolution, high temporal resolution, high spectral resolution, wide observation range, and low cost. We used 240 MODIS images on GEE.

**Table 1.** MOD13Q1 and HJ-1A/B dataset description.

| Dataset | Nominal Resolution | Temporal Granularity | Temporal Coverage |
|---------|--------------------|----------------------|--------------------|
| MOD13Q1 | 250 m | 16 day | 2010.6–2019.9 |
| HJ-1A/B CCD | 30 m | 4 day | 2010.6–2019.9 |

2.2.2. HJ-1A/B

The verification data for the disaster monitoring in this study were the HJ-1A/B data with a 30-m spatial resolution from the environmental disaster mitigation satellite (Table 1). The Chinese HJ-1A/B satellite makes synchronous ground observations, the charge-coupled device (CCD) sensor captures the ground features with a 30-m pixel resolution at a minimum angle, and four bands cover the visible light and near-infrared wavelength ranges. Each satellite has two CCD sensors, and the constellation consisting of two satellites forms an observation network covering China and its surrounding areas, featuring large-scale, all-weather, all-day, dynamic environmental and disaster monitoring capabilities. In addition, it was combined with the crop insurance plots from 2010–2019 in order to determine the disaster scope through the ground measured data. Insurance company personnel carried out a field verification on 80% of the plots, and the accuracy was determined to be >95%. These data were used to validate the extraction extent of the disasters in this study.

2.2.3. Meteorological Data

The spatial and temporal distribution characteristics of agricultural disasters in Heilongjiang Province and its prefecture-level cities from 2010–2019 were analyzed based on the precipitation, temperature, humidity, and sunshine duration meteorological data gathered by the Heilongjiang

Provincial Bureau of Statistics (http://www.hlj.stats.gov.cn/) and the Yearbook of meteorological disasters in China.

### 2.2.4. Cultivated Land Range Data

In this study, in order to avoid the influence of other land types and to conduct phenological zoning for the construction of the disaster monitoring model, the disaster monitoring and extraction for the cultivated land was performed using the land range extracted from the global 30-m land cover data. The land use classification data were from the Northeast Institute of Geography and Agroecology of the Chinese Academy of Sciences, which used 2014 land samples. Taking the CCD images from the China Resources No. 1 satellite and the Landsat remote sensing images as the main data sources, and adopting the manual visual interpretation method, we obtained the cultivated land range, as shown in Figure 2.

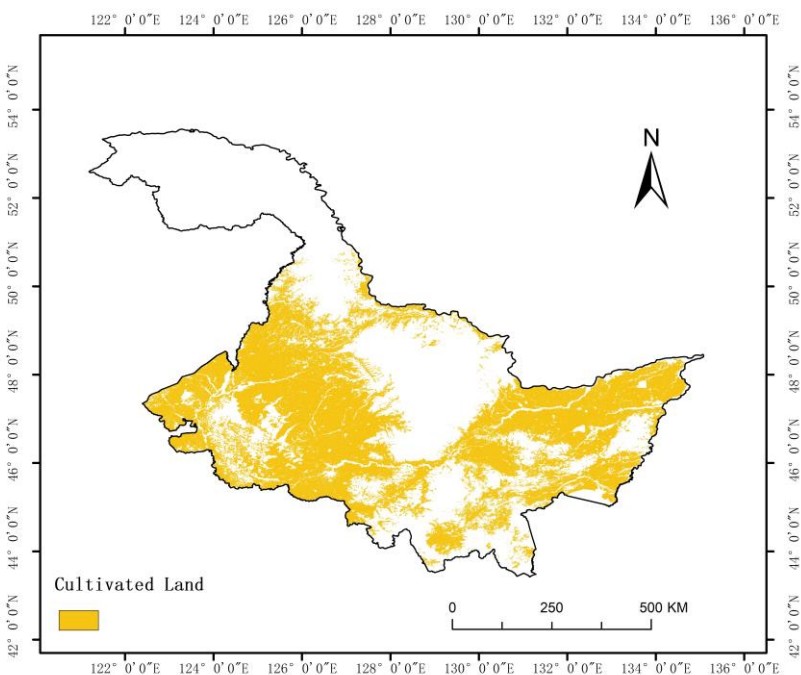

**Figure 2.** Cultivated land of Heilongjiang Province.

### 2.3. Methods

#### 2.3.1. Data Preprocessing

Here, the MOD13Q1 reflectivity product was used to construct the model based on GEE. GEE contains over 200 public datasets and more than 5 million images, and is increasing at a rate of approximately 4000 images per day. Images uploaded into GEE are preprocessed. In GEE, the MOD13Q1 NDVI products are calculated based on atmosphere-corrected bidirectional surface reflectance, which is shielded against water, clouds, heavy aerosols, and cloud shadows. We selected the good data and marginal data from the SummaryQA (Table 2) in order to remove the impact of clouds and snow and ensure that the extracted disaster scope was not affected by outliers. HJ-1A/B images have undergone radiometric calibration, atmospheric correction, and geometric correction.

**Table 2.** Description of Summary QA in MOD13Q1.

| Band | Bits 0-1: VI Quality (MODLAND QA Bits) |
|------|-----------------------------------------|
| SummaryQA Bitmask | 0: Good data, use with confidence<br>1: Marginal data, useful but look at detailed QA for more information<br>2: Pixel covered with snow/ice<br>3: Pixel is cloudy |

### 2.3.2. Phenological Remote Sensing Zoning Method

In terms of the remote sensing image processing, 23 MODIS (MOD13Q1) remote sensing data with a spatial resolution of 250 m synthesized over 16 days in 2014 were employed to extract 11 crop phenological features, and the multi-phase NDVI time series was smoothly reconstructed using Savitzky–Golay filtering. The dynamic threshold method was utilized to extract the key phenological values, and the intervention of different vegetation types and soil background values were eliminated. Based on the time series changes of the NDVI curve, the beginning of the crop growth period was defined as the sharp rise of the NDVI curve on the left side, i.e., the time when the increasing range was 20% of the overall increase. At the same time, the end of the crop growth period was defined as the sharp decrease of the NDVI curve, i.e., the time when the decreasing range was 20% of the overall increase. The 11 extracted phenological characteristic parameters are listed in Table 3. The regions with similar phenological values were categorized as a single study area, and multiscale segmentation was carried out on the cultivated land. Via this method, different crops with various geographical distributions and growth conditions were divided into different agricultural phenological zones. It was required that the laws of zonality and non-zonality for phenological distribution as well as the principle of crop similarity and difference be followed, and certain zoning methods were adopted in order to divide a region into units of different grades, with clear differences in crop growth. Pursuant to the method described above, Heilongjiang Province was divided into 39 phenological areas [59].

**Table 3.** Definition of phenological parameters in remote sensing.

| Name | Definition Interpretation |
|------|---------------------------|
| $NDVI_{Start}$ | Start of crop growth period |
| $NDVI_{End}$ | End of crop growth period |
| $NDVI_{Amp}$ | Amplitude |
| $NDVI_{Base}$ | Average of NDVI at start and end |
| $NDVI_{Length}$ | Length of crop growth period |
| $NDVI_{Small}$ | Integral of the average NDVI for the entire period |
| $NDVI_{Max}$ | NDVI maximum |
| $NDVI_{Left}$ | Slope between the 20% and 80% amplitude points on the left side of the rising curve |
| $NDVI_{Right}$ | Slope between the 20% and 80% amplitude on the right side of the descending curve |
| $NDVI_{Mid}$ | Midpoint of the entire period |
| $NDVI_{Large}$ | NDVI integral for the entire period |

### 2.3.3. Construction of Three Disaster Monitoring Models

Certain differences exist in Heilongjiang Province: the phenological periods and cultivated land planting structures, the vegetation indices of crops growing at the same time but in different areas, and the vegetation indices of different crops. Therefore, the results of disaster range recognition and extraction based directly on the NDVI value difference of a certain phase are not precise and not universal. For this situation, the following three models were proposed and calculated in GEE:

The $R_{NDVI\_TM(i)}$ model with normalized difference median vegetation index time:

$$R_{NDVI\_TM(i)} = \frac{NDVI_{(i)} - NDVI_{TMED(i)}}{NDVI_{TMED(i)}} \times 100\% \tag{1}$$

where $R_{NDVI\_TM(i)}$ represents the time standardization value of the $NDVI_{(i)}$ of the ith time phase in a certain year, $NDVI_{(i)}$ is the NDVI value of the *i*th time phase in a certain year, and $NDVI_{TMED(i)}$ is the NDVI value of the *i*th time phase for five consecutive years. The smaller the $R_{NDVI\_TM(i)}$ value, the less the vegetation grows. Five years was selected as the time scale because longer time scales are susceptible to management decisions such as dryland diversion, crop rotation, and changes of planting structure. Meanwhile, shorter time scales cannot reflect the time trend and are prone to the influence of individual annual outliers.

The $R_{NDVI\_AM(i)(j)}$ model of phenology standardization of the median value of the normalized difference vegetation index is:

$$R_{NDVI\_AM(i)(j)} = \frac{NDVI_{(i)} - NDVI_{AMED(i)(j)}}{NDVI_{AMED(i)(j)}} \times 100\% \tag{2}$$

where $R_{NDVI\_AM(i)(j)}$ is the phenological standardization value of the median *NDVI(i)* in the *j*th phenological region of the *i*th phase in a certain year, *NDVI(i)* Is the NDVI value of the *i*th phase in a certain year, and $NDVI_{AMED(i)(j)}$ is the median value of the NDVI region in the *j*th phenological region of the *i*th phase in a certain year. The smaller the $R_{NDVI\_AM(i)(j)}$ value, the worse the vegetation grows.

The $R_{NDVI\_ZM(i)(j)}$ model is based on an improvement of Equations (1) and (2). Given that the median value curve of the NDVI region for the same phenological area in different years may be affected by the change of crop planting structure and other factors, the median NDVI values extracted at the same time in different years can exhibit great differences. Therefore, the regional median of the phase NDVI of phase I for five consecutive years is proposed as an alternative.

$$R_{NDVI\_ZM(i)(j)} = \frac{NDVI_{(i)} - NDVI_{ZWED(i)(j)}}{NDVI_{ZMED(i)(j)}} \times 100\% \tag{3}$$

where $R_{NDVI\_ZM(i)(j)}$ is the spatiotemporal standardization value of the median *NDVI(i)* in the *j*th phenological region of the *i*th phase in a certain year, *NDVI(i)* is the NDVI value of the *i*th phase in a certain year, and $NDVI_{ZMED(i)(j)}$ is the standardized median value of the NDVI in the *j*th phenological region of the *i*th phase for five consecutive years. The smaller the $R_{NDVI\_ZM(i)(j)}$ value, the less the vegetation grows.

### 2.3.4. Determination of Threshold Value

The phenological period of crops in Heilongjiang Province is relatively late, and the phenological periods of the three major crops in Heilongjiang Province are shown in Table 4.

From mid-April to early June, crops in Heilongjiang Province are in the seeding stage and seedling stage, during which the crop coverage is low and the NDVI value is small, and thus images are easily susceptible to the soil background value. Therefore, this study began extracting the disaster scope from the day-of-year (DOY) 177. In mid-September, precocity occurs in some crops, so the disaster area cannot be directly extracted on DOY 273. In this study, images between DOY 161 and DOY 257 were selected. A total of 113 typical disasters reported by insurance companies from 2011 to 2019 were chosen as sample data. Otsu is a method to select threshold from Gray-Level histograms. Otsu's method was employed to determine the appropriate threshold value for extracting the disaster scope and verifying its universal applicability via the GEE monitoring model. We adopted the average value without the extreme outliers as the threshold in order to distinguish between disasters and non-disasters and calculated the proportions of the MODIS image extraction results and the insured plots to obtain the corresponding error size and verify its accuracy.

**Table 4.** Phenological periods of main crops in Heilongjiang Province.

| Crop Species | Crop Phenology | | (10 Days/Month) | | | |
|---|---|---|---|---|---|---|
| Rice | Sowing and seedling raising<br>Mid-April–mid-May | Transplanting and rejuvenation<br>Late May–early June | Tillering<br>Mid-June–mid-July | Booting and tasseling<br>Late July–mid-August | Milk<br>Late August–early September | Mature<br>Mid-September–late September |
| Corn | Seed and emergence<br>Late April–early May | Seedling<br>Mid-May–mid-June | Jointing<br>Late June–mid-July | Emasculation<br>Late July–early August | Milk<br>Mid-August–early September | Mature<br>Mid-September–late September |
| Soybean | Seed and emergence<br>Early May–late May | Third Leaf<br>Early June–late June | Parabranching<br>Late June | Flowering<br>Early July–mid July | Podding<br>Mid-August–early September | Mature<br>Mid-September–late September |

### 2.3.5. Disaster Extraction

When crops suffer from disasters, the values of $R_{NDVI\_TM(i)}$, $R_{NDVI\_AM(i)(j)}$, and $R_{NDVI\_ZM(i)(j)}$ are slightly lower than their normal levels. Therefore, when the standardized value of a certain regional model was found to be less than a threshold value, the crop was identified as being affected by a disaster. The smaller the values of $R_{NDVI\_TM(i)}$, $R_{NDVI\_AM(i)(j)}$, and $R_{NDVI\_ZM(i)(j)}$, the more severe the damage. Thus, this study analyzed the $R_{NDVI\_TM(i)}$, $R_{NDVI\_AM(i)(j)}$, and $R_{NDVI\_AM(i)(j)}$ values in Heilongjiang Province from 2010–2019 in accordance with the time sequence. The average value extracted using Otsu's method was taken as the threshold value, and the disaster scope was extracted from the corresponding remote sensing disaster monitoring model via the determined threshold values of each time phase. Given the spatial resolution of the MODIS data and the need to remove small patches after the extraction of agricultural disasters, the disaster areas covering <6 pixels (approximately 40 hectares) were eliminated in order to obtain the agricultural disaster scope of Heilongjiang Province from 2010–2019.

### 2.3.6. Accuracy Verification

In order to test the accuracy of the disaster ranges extracted by the three types of disaster monitoring models, the actual data from surface disasters during crop growth periods from 2010–2019 were selected as the validation samples. The verification data consisted of the disaster area extracted by the insurance company based on HJ-1A/B CCD images and field sampling. Our default assumption was that this was the true value, i.e., the extracted disaster range was identified as the actual disaster range, which was then used to verify the accuracy of the disaster range in the MODIS image area. The spatial resolution of the HJ-1A/B images is 30 m, and the accuracy of the disaster range extraction is higher than that of the MODIS images. Therefore, the disaster range extracted from the image was taken as the real value.

We took Absolute error = | Extracted value – True value |, i.e., the Absolute value between the disaster result extracted from the MODIS data and the disaster result extracted from HJ-1A/B, as the accuracy evaluation parameter. Finally, the errors of 285 verification samples for the different models were calculated as the average values of the accuracy test.

## 3. Results

### 3.1. Phenological Division of Cultivated Land

The purpose of utilizing the key phenological values as the basis of zoning in Heilongjiang Province was to combine the regions with similar phenological values into a single study region, then conduct multiscale segmentation within the cultivated land. After conducting numerous experiments and using the average segmentation evaluation index (ASEI) for calculation and analysis, we discovered that the ASEI value reached its maximum when the optimal segmentation scale was 70. The 39 phenological regions that were ultimately obtained are shown in Figure 3. After the cultivated land was categorized into regions according to its phenological values, the median values of the different phenological regions were extracted from the processed images in GEE as $NDVI_{AMED(i)(j)}$ and $NDVI_{ZMED(i)(j)}$, and $R_{NDVI\_AM(i)(j)}$ and $R_{NDVI\_ZM(i)(j)}$ were then calculated.

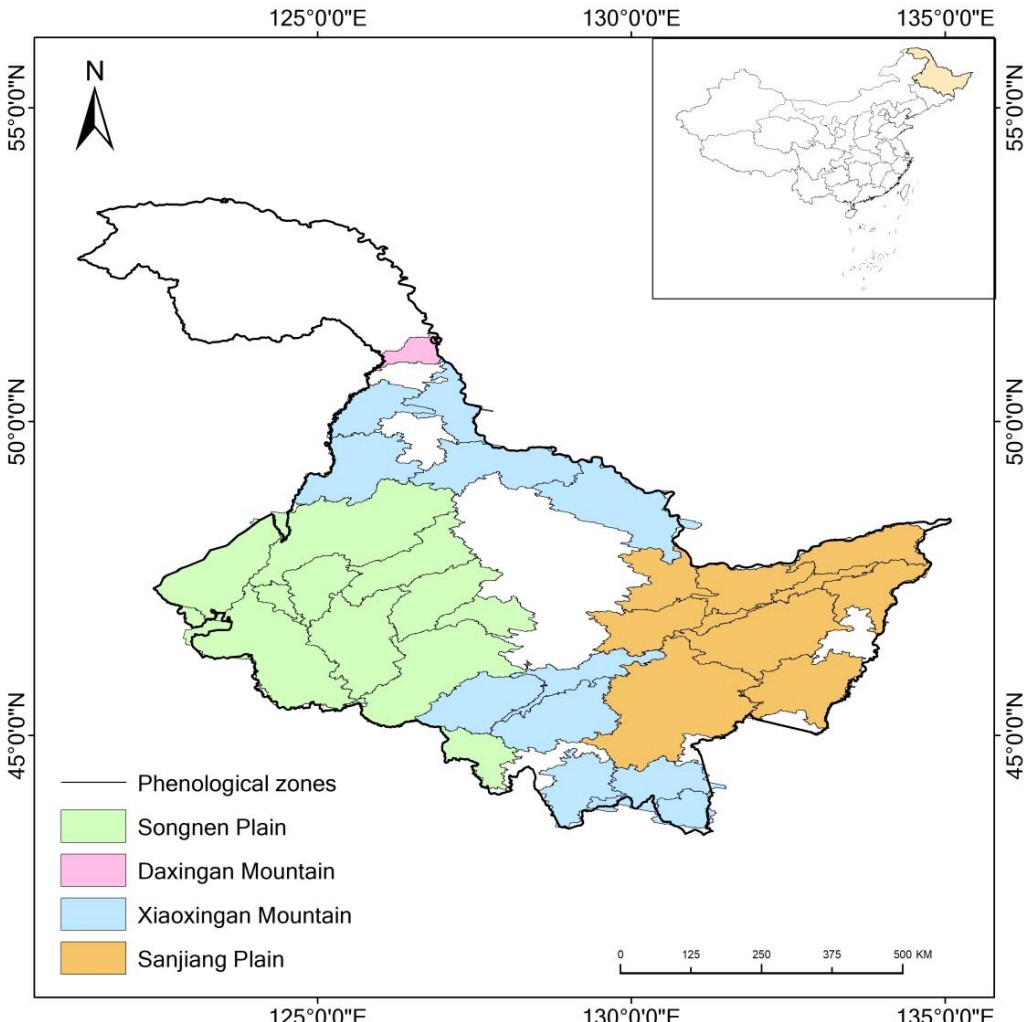

**Figure 3.** Phenological zones in Heilongjiang Province.

*3.2. Precision Analysis*

We used Otsu's method to extract the threshold value of DOY 113 sample points of different disaster types in GEE. From Table 5, we find that in the three models, the threshold size was mainly distributed between −0.1 and −0.2. The thresholds extracted from different disaster types and by different models differed from one another. The error in the Table 5 was determined based on the difference between the proportion of disaster results extracted by the HJ-1A/B images provided by the insurance company and the proportion of disaster results extracted by MODIS images. Among them, the errors of insect and wind disasters were larger. At the same time, we extracted the threshold values of insect and wind disasters after 16 days and conducted a precision analysis. It was found that the errors of the results of these two disasters were smaller and the accuracy was higher from images observed after 16 days. Therefore, we used the images from 16 days later to calculate the threshold values of insect and wind disasters. Since the errors of hailstorms, drought disasters, and flood disasters were small, the MODIS image close to the time of a given disaster was used to calculate the disaster threshold for disaster monitoring.

**Table 5.** Otsu extraction disaster threshold and disaster error analysis example.

| Model | Definition Interpretation | Proportion of HJ-1A/B Monitoring Results for the Insured Land (%) | Threshold | Proportion of MODIS Monitoring Results for the Insured Land (%) | Error (%) |
|---|---|---|---|---|---|
| $R_{NDVI\_TM(i)}$ | 20170803Youyi hailstorm | 1.31 | −0.15 | 1.39 | 0.08 |
| | 20180703Tonghe flood | 2.27 | −0.16 | 2.36 | 0.09 |
| | 20160813Longjiang drought | 0.47 | −0.14 | 0.57 | 0.10 |
| | 20170802Fuyuan flood | 0.17 | −0.08 | 0.30 | 0.13 |
| | 20180703Zhaodong flood | 13.14 | −0.11 | 12.71 | 0.44 |
| | 20120702Maqiaohe hailstorm | 84.82 | −0.14 | 85.50 | 0.68 |
| | 20160702Hailun hailstorm | 4.60 | −0.15 | 3.88 | 0.73 |
| | 20160829Gannan drought | 1.47 | −0.16 | 0.66 | 0.82 |
| | 2018080Luobei wind hazard | 2.47 | −0.16 | 0.25 | 2.22 |
| | 20170901Beian wind hazard | 16.63 | −0.10 | 35.04 | 18.41 |
| $R_{NDVI\_AM(i)(j)}$ | 20170803Youyi hailstorm | 6.87 | −0.10 | 10.96 | 4.09 |
| | 20180703Tonghe flood | 2.27 | −0.14 | 3.21 | 0.94 |
| | 20160813Longjiang drought | 21.68 | −0.14 | 25.74 | 4.06 |
| | 20170802Fuyuan flood | 0.87 | −0.17 | 0.77 | 0.10 |
| | 20180703Zhaodong flood | 23.19 | −0.14 | 19.32 | 3.88 |
| | 20120702Maqiaohe hailstorm | 84.82 | −0.17 | 89.06 | 4.24 |
| | 20160702Hailun hailstorm | 54.57 | −0.18 | 30.62 | 23.95 |
| | 20160829Gannan drought | 1.47 | −0.15 | 2.43 | 0.95 |
| | 2018080Luobei wind hazard | 2.47 | −0.15 | 0.63 | 1.84 |
| $R_{NDVI\_ZM(i)(j)}$ | 20170803Youyi hailstorm | 38.41 | −0.13 | 51.15 | 12.73 |
| | 20180703Tonghe flood | 6.87 | −0.15 | 8.22 | 1.35 |
| | 20160813Longjiang drought | 2.27 | −0.14 | 1.79 | 0.48 |
| | 20170802Fuyuan flood | 41.16 | −0.14 | 63.13 | 21.98 |
| | 20180703Zhaodong flood | 0.87 | −0.13 | 0.97 | 0.10 |
| | 20120702Maqiaohe hailstorm | 13.14 | −0.11 | 17.28 | 4.14 |
| | 20160702Hailun hailstorm | 84.82 | −0.16 | 91.64 | 6.82 |
| | 20160829Gannan drought | 36.25 | −0.16 | 36.43 | 0.18 |
| | 2018080Luobei wind hazard | 3.01 | −0.16 | 3.74 | 0.73 |
| | 20170901Beian wind hazard | 2.47 | −0.18 | 0.47 | 1.99 |

After removing the extreme values from the thresholds of the different phases in the three monitoring models, the mean value was taken as the threshold value of the time phase. The sizes and errors of the average thresholds are listed in Table 6. Generally speaking, as time increased, the threshold values increased, indicating that the disasters across the entire province exhibited a gradual decreasing trend during the crop growth period. In addition, the difference of the threshold value between the $R_{NDVI\_AM(i)(j)}$ and $R_{NDVI\_ZM(i)(j)}$ models for the same time phase was small, implying that the extraction disaster scopes may have been similar. At the same time, as shown in Tables 7 and 8, based on either the environmental star monitoring results or the MODIS monitoring results, the proportion of hailstorms was the highest, followed by drought and flood disasters. Windstorms and insect disasters often accounted for a small proportion of the monitoring results for the insured land, i.e., the disaster areas resulting from these events were small. On DOY 209, the relatively large average error may have led to the large error of the disaster area extracted during this period. Having selected the DOY 285 samples to test the accuracy of the three monitoring models, our calculations revealed that the average precision values of the $R_{NDVI\_TM(i)}$, $R_{NDVI\_AM(i)(j)}$, and $R_{NDVI\_ZM(i)(j)}$ monitoring models were 97.46%, 96.90%, and 96.67%, respectively. In Table 8, the average errors of droughts, windstorms, hailstorms, and floods were smaller and their accuracy values were higher, while the average error of insect infestation was larger and its accuracy was lower.

**Table 6.** Mean threshold and error analysis of each phase.

| Model | DOY | Threshold | Average Error (%) |
|---|---|---|---|
| $R_{NDVI\_TM(i)}$ | 177 | −0.13 | 2.90 |
| | 193 | −0.16 | 7.78 |
| | 209 | −0.15 | 6.29 |
| | 225 | −0.15 | 4.22 |
| | 241 | −0.13 | 4.58 |
| | 257 | −0.14 | 2.83 |
| $R_{NDVI\_AM(i)(j)}$ | 177 | −0.15 | 5.89 |
| | 193 | −0.15 | 3.70 |
| | 209 | −0.15 | 7.51 |
| | 225 | −0.13 | 4.99 |
| | 241 | −0.13 | 5.11 |
| | 257 | −0.13 | 7.08 |
| $R_{NDVI\_ZM(i)(j)}$ | 177 | −0.16 | 5.27 |
| | 193 | −0.16 | 4.32 |
| | 209 | −0.15 | 7.44 |
| | 225 | −0.13 | 5.31 |
| | 241 | −0.15 | 3.16 |
| | 257 | −0.13 | 4.06 |

**Table 7.** Sample comparison table for the accuracy test of MODIS data disaster range extraction based on HJ-1A/B CCD image.

| Model. | Definition Interpretation | Proportion of HJ-1A/B Monitoring Results for the Insured Land (%) | Threshold | Proportion of MODIS Monitoring Results for the Insured Land (%) | Error (%) |
|---|---|---|---|---|---|
| $R_{NDVI\_TM(i)}$ | 20180801Tongjiang flood | 7.08 | −0.15 | 8.44 | 1.36 |
| | 20180803Tonghe wind hazard | 3.41 | −0.15 | 3.61 | 0.20 |
| | 20180803Suiling wind hazard | 2.62 | −0.15 | 1.79 | 0.83 |
| | 20160829Nehe drought | 5.60 | −0.13 | 0.86 | 4.73 |
| | 20120914Hulan Insect | 20.36 | −0.14 | 20.90 | 0.54 |
| | 20120829Wuchang Insect | 8.79 | −0.13 | 0.14 | 8.65 |
| | 2017090Nenjiang flood | 12.35 | −0.14 | 16.77 | 4.42 |
| | 20180901Zhaodong hailstorm | 51.82 | −0.14 | 58.66 | 6.84 |
| | 20180901Hailun hailstorm | 52.13 | −0.14 | 69.87 | 17.74 |
| | 20190907Nehe flood | 22.34 | −0.14 | 28.85 | 6.51 |
| $R_{NDVI\_AM(i)(j)}$ | 20180801Tongjiang flood | 4.10 | −0.13 | 4.28 | 0.18 |
| | 20180803Tonghe wind hazard | 8.55 | −0.13 | 10.98 | 2.43 |
| | 20180803Suiling wind hazard | 2.62 | −0.13 | 3.22 | 0.60 |
| | 20160829Nehe drought | 1.92 | −0.13 | 1.06 | 0.86 |
| | 20120914Hulan Insect | 20.36 | −0.13 | 4.64 | 15.72 |
| | 20120829Wuchang Insect | 8.79 | −0.13 | 0.69 | 8.11 |
| | 20170901Nenjiang flood | 6.94 | −0.13 | 13.73 | 6.79 |
| | 20180901Zhaodong hailstorm | 67.07 | −0.13 | 70.06 | 2.98 |
| | 20180901Hailun hailstorm | 80.72 | −0.13 | 92.93 | 12.21 |
| | 20190907Nehe flood | 50.25 | −0.13 | 48.16 | 2.09 |
| $R_{NDVI\_ZM(i)(j)}$ | 20180801Tongjiang flood | 4.10 | −0.13 | 4.73 | 0.63 |
| | 20180803Tonghe wind hazard | 8.55 | −0.13 | 10.98 | 2.43 |
| | 20180803Suiling wind hazard | 2.62 | −0.13 | 3.22 | 0.60 |
| | 20160829Nehe drought | 1.92 | −0.13 | 1.18 | 0.74 |
| | 20120914Hulan Insect | 20.36 | −0.13 | 1.80 | 18.55 |
| | 20120829Wuchang Insect | 8.79 | −0.13 | 12.41 | 3.62 |
| | 20170901Nenjiang flood | 6.94 | −0.13 | 8.53 | 1.58 |
| | 20180901Zhaodong hailstorm | 67.07 | −0.13 | 62.44 | 4.63 |
| | 20180901Hailun hailstorm | 80.72 | −0.13 | 91.63 | 10.90 |
| | 20190907Nehe flood | 50.25 | −0.13 | 42.80 | 7.45 |

**Table 8.** Average errors of the three monitoring models for different disasters (%).

| | $R_{NDVI\_TM(i)}$ | $R_{NDVI\_AM(i)(j)}$ | $R_{NDVI\_ZM(i)(j)}$ |
|---|---|---|---|
| Hailstorm | 3.16 | 2.93 | 3.52 |
| Pest plague | 6.70 | 11.91 | 12.33 |
| Wind hazard | 1.61 | 2.28 | 1.77 |
| Drought | 4.91 | 1.68 | 5.39 |
| Flood | 2.48 | 2.85 | 2.94 |

### 3.3. Consistency Analysis of the Applicability and Extraction Scope of Different Models

Based on the thresholds of the different time phases listed in Table 6, the typical disasters verified by the HJ-1A/B monitoring range and the disaster scope of Heilongjiang Province from 2010–2019 were extracted. These results are presented in Figures 4 and 5.

As shown in Table 8, the average errors of the hailstorm and wind disasters extracted by the $R_{NDVI\_TM(i)}$ and $R_{NDVI\_ZM(i)(j)}$ models were relatively small, and the disaster extraction ranges of these models for the actual observations shown in Figure 5 were similar. The average flood disaster errors extracted by the $R_{NDVI\_AM(i)(j)}$ and $R_{NDVI\_ZM(i)(j)}$ were small, and the disaster extraction ranges of these models for the actual observations were similar. In terms of drought, however, although the error difference between the $R_{NDVI\_TM(i)}$ and $R_{NDVI\_ZM(i)(j)}$ was smaller, the disaster range extracted by the $R_{NDVI\_AM(i)(j)}$ was similar to that extracted by the $R_{NDVI\_ZM(i)(j)}$ for the actual observations.

The crops ripen once a year in Heilongjiang Province, although the three models monitored and extracted the disaster areas from mid-June to mid-September with little difference. It can be seen from Table 9 that the three monitoring models exhibited similar ratios of phase disaster range to the cultivated land range across the entire province during the period DOY 177–DOY 225, among which the $R_{NDVI\_TM(i)}$ and $R_{NDVI\_ZM(i)(j)}$ displayed a small difference in this ratio on DOY 177. Figure 4 shows that their extracted disaster ranges were also relatively close. For the DOY 117–DOY 209 phases, there was a small difference between the $R_{NDVI\_AM(i)(j)}$ and $R_{NDVI\_ZM(i)(j)}$ in the disaster scope proportion of the cultivated land across the entire province, and the disaster scopes extracted in Figure 4 were more consistent. In phases DOY 241–DOY 257, the $R_{NDVI\_TM(i)}$ and the other two monitoring models indicated that the extracted disaster range accounted for a larger percentage of the total cultivated land area in the province, and the extracted disaster range exhibited a larger difference. The main reason for this finding is that in DOY 241–DOY 257, the $R_{NDVI\_AM(i)(j)}$ and $R_{NDVI\_ZM(i)(j)}$ were more sensitive to waterlogging, resulting in a larger monitored range.

**Table 9.** Ratio of 2017 disaster scope to cultivated land area in Heilongjiang Province (%)

| DOY. | $R_{NDVI\_TM(i)}$ | $R_{NDVI\_AM(i)(j)}$ | $R_{NDVI\_ZM(i)(j)}$ |
|------|------|------|------|
| 177 | 11.29 | 13.17 | 11.83 |
| 193 | 6.04 | 7.22 | 6.78 |
| 209 | 3.03 | 4.41 | 4.06 |
| 225 | 3.17 | 4.38 | 3.97 |
| 241 | 7.96 | 10.30 | 11.58 |
| 257 | 15.43 | 18.59 | 16.85 |

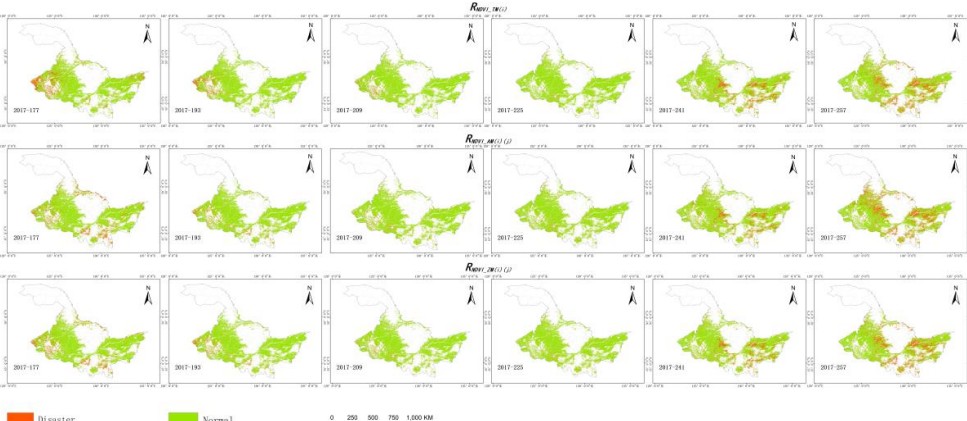

**Figure 4.** 2017 disaster distribution maps of Heilongjiang Province for the three monitoring models.

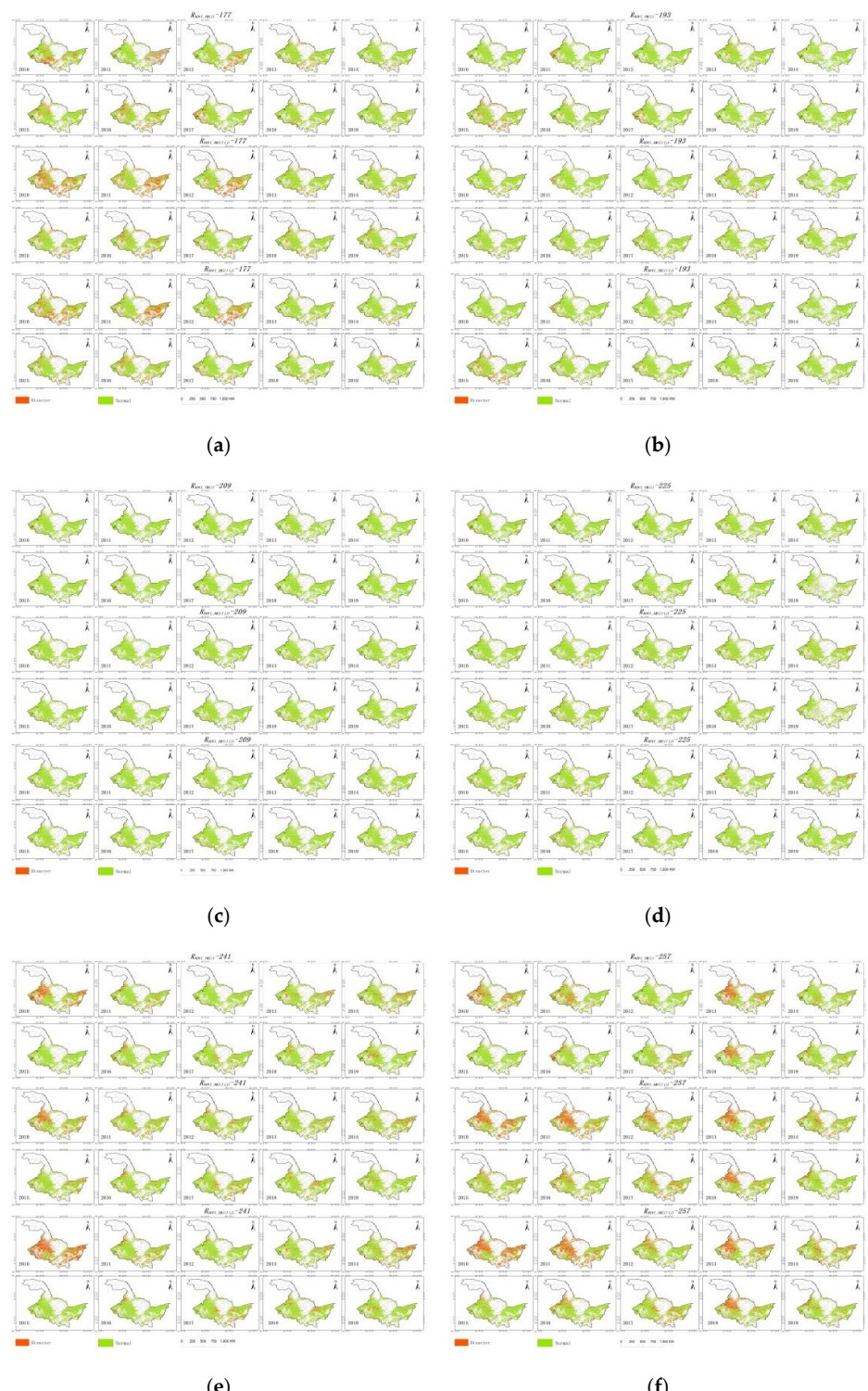

**Figure 5.** Disaster distribution maps of Heilongjiang Province during different phases from 2010–2019:
(**a**) DOY 177, (**b**) DOY 193, (**c**) DOY 209, (**d**) DOY 225, (**e**) DOY 241, and (**f**) DOY 257.

*3.4. Analysis of Spatiotemporal Patterns of Disasters in the Study Area*

3.4.1. Spatial and Temporal Pattern Analysis of 2017 Disasters in Heilongjiang Province

After extracting the disaster area of Heilongjiang Province using the threshold values of different
time phases, the spatial and temporal distributions of disasters in Heilongjiang Province over the past

10 years could be analyzed in combination with the corresponding meteorological data. Taking 2017 as an example, as shown in Figures 4 and A1 (see Appendix A), disasters in June were concentrated in the western and southeastern regions. Longjiang County and Tailai County had little rain over the years, and drought occurred frequently. In early July, the crop situation improved, although serious disasters still occurred in southern areas such as Wuchang due to heavy rain, as well as in Keshan County, Nehe, and other areas. In late July, the crops grew well, and the disasters were concentrated in the western and northwestern areas of Heilongjiang Province, while the southeastern area of Fuyu County had a low vegetation index for the entire month of July, and disasters occurred. In mid-August, the vegetation growth in the Jiamusi and Suihua areas was poor, and the trend worsened in early September.

In terms of the spatial and temporal distribution trends of disasters, based on distribution consistency, the time period DOY 177–DOY 193 was labeled time period 1, and DOY 193–DOY 209, DOY 209–DOY 225, and DOY 225–DOY 257 were designated period 2, 3, and 4, respectively. The time variation characteristics of the provincial disasters revealed that the disaster area exhibited a downward trend from period 1 to period 2, and this trend continued to period 3, when it reached its minimum. Entering period 4, however, the disaster area increased rapidly, which was consistent with the disaster area change of the insured land across the entire province. In period 1, the disasters were mainly distributed in the west and south, where Qiqihar, Heihe, Daqing, Mudanjiang, Anda, and Wudalianchi were severely affected, and the level of severity gradually decreased with time. During the second period, agricultural disasters were mainly concentrated in Qiqihar, Heihe (the Aihui District, Nenjiang County, Xunke County, Sunwu County, and Wudalianchi), Suihua, Nehe, and other locations, all of which were severely impacted. In the third period, the disasters mainly occurred in the west, south, and central portions of Heilongjiang Province. Qiqihar and Heihe were still the most affected areas; Jixi, Jiamusi, and Suihua were clearly stricken; and Duerbert, Zhaozhou, and Acheng also suffered severely. During period 4, crops in many areas had already entered the harvest season by late August and early September, particularly rice, which is grown widely in the Sanjiang region of the Jiamusi belt, leading to a significant increase of the disaster area in the Kiamusze region, as seen in Figure 4. This gave the impression that agricultural disasters in Heilongjiang Province were concentrated in the northeast. In summary, the 2017 agricultural disasters in Heilongjiang Province were mainly concentrated in the northeast, south, west, and central regions.

3.4.2. Spatial and Temporal Disaster Pattern Analysis of Different Phases in The Study Area from 2010 to 2019

Comparing the same time phase images of different years allows for a deeper analysis of the spatial and temporal pattern distributions of disasters.

The analysis of the disaster range and meteorological data over the 10-year study period revealed that on DOY 177 in 2010, 2011, 2012, 2016, and 2017 a large range of disasters occurred. In 2010, Heilongjiang Province continued to experience low temperatures in the winter and spring. The weather warmed late and the soil defrosted slowly. In May, precipitation was unusually heavy, leading to late field seeding. Therefore, the bare soil area was extensive, resulting in the large 2010 disaster scope shown in Figure 5a. Severe convective weather generated a hailstorm in the Beilin District of Suihua, Hailun, Lanxi County, Qingan County, Suiangxian County, and the Hulan District of Harbin. The actual range of the hailstorm was consistent with the ranges extracted from the three models. Due to the sustained high temperatures and sparse rainfall from late May through June, parts of the Songnen Plain, the northern forest region, the northern Sanjiang Plain, and Mudanjiang experienced drought conditions. The drought-stricken areas were mainly distributed in the Greater Hinggan Mountains and the Mudan River region. In the Mudan River region, the three monitoring models were consistent. In the forest regions, however, the $R_{NDVI\_AM(i)(j)}$ and $R_{NDVI\_ZM(i)(j)}$ were more sensitive to drought monitoring, resulting in more extensive drought extraction in the northern areas. In June 2011, rainstorm and flood disasters occurred in Heilongjiang Province, severely impacting Fujin, Qiqihar, and other areas. In addition, hailstorms occurred in many locations. Figure 5a reveals that the disaster area extracted in

2011 was concentrated in the western and northeastern sections of Heilongjiang Province, which was consistent with the meteorological data. In June 2012, precipitation in the eastern part of Harbin and the Sanjiang Plain continued to be low, eventually resulting in drought. Shuangyashan, Baoqing, Wuchang, Tonghe, Fangzheng, and other counties suffered from severe drought. A hailstorm occurred in Qiqihar Mountain County. In Figure 5a, the disaster areas extracted in 2012 were concentrated in the eastern and western sections of Heilongjiang Province, and the extraction of drought areas was good. In June 2014, strong convective weather occurred in some areas of Heilongjiang Province. Windstorms and hailstorms occurred with high frequency, impacting a wide area and resulting in severe losses. The extraction process revealed that the disasters were concentrated in the Jiamusi area, which is in the southern part of the province, and Suihua, which is in the western part. In June 2015, strong convective weather occurred in Heilongjiang Province, with a high frequency of hailstorms. The extracted disasters were concentrated in the northwestern, northeastern, and southern sections of Heilongjiang Province. In June 2016, there was a large amount of precipitation in the province, with heavy rain concentrated in most of the Songnen Plain and the northern portion of the Sanjiang Plain. Yanshou County and other areas suffered from severe waterlogging due to the heavy rainfall, and this meteorological disaster was consistent with the extracted disaster in this county. In mid-June 2017, rainstorms and floods occurred frequently, and waterlogging was severe in Nehe and other locations, which was consistent with the monitoring results.

During the time phase DOY 193–DOY 209, the disasters occurring in 2012, 2015, 2016, and 2017 were relatively serious. In 2010, the average rainfall of Heilongjiang Province in this phase was higher than the average of a normal year. The rainstorms and floods in July damaged 221,000 hectares of crops. Figure 5b shows that in 2010 floods mainly occurred in Heihe, Suihua, and Harbin. From May to mid-July 2012, rainfall in the eastern part of Harbin and the Sanjiang Plain continued to be low, causing moderate meteorological drought, including severe drought in the Shuangyashan urban area, as well as Baoqing, Fuchang, Tonghe, Fangzheng, and other counties. At the end of July, Daqing and many other cities suffered from severe flooding and waterlogging disasters, which was consistent with the disaster extraction range. In addition, there were mild disasters in the central and northern regions of the extraction range. In July 2013, heavy rainfall occurred in Heilongjiang Province, causing regional floods along the Heilongjiang, Nenjiang, and Songhua rivers. The disaster distribution map clearly shows that greater waterlogging occurred along these rivers. Strong convective weather was observed in some areas of Heilongjiang Province. At the end of July, hailstones pummeled the Beilin District of Suihua, which was consistent with the extracted disaster area. In July 2014, Jiamusi was hit by severe hailstorms, which was also consistent with the extracted disaster scope. Meanwhile, according to the extracted disaster map, the entire province was flooded and waterlogging was serious during this period. In 2015, Heilongjiang Province witnessed frequent rainstorms and floods, and severe convective weather occurred in many areas. For example, the Hulan District of Harbin was hit by tornadoes and hail. In mid-July, Hulin experienced a rainstorm, which coincided with the disaster area extracted on DOY 209. In addition, there were a few disasters in the northeastern portion of Heilongjiang Province. In July 2016, the continuous high temperatures and low rainfall in the province led to a drought on the western Songnen Plain in mid-July. Rainstorms and floods occurred frequently, especially in late July, mainly in most sections of the Songnen Plain and the northern Sanjiang Plain. These events were all consistent with the extraction disaster scope. In addition, there was a small disaster in the northwestern part of the province on DOY 209. In July 2017, the average temperatures were excessive, causing most of the Songnen Plain to become arid. In the middle of the year, the western region suffered from a continuous drought due to insufficient precipitation. By the end of July, Duerbert, Zhaozhou, Zhaoyuan, and Acheng were experiencing drought conditions as well. Heavy rain and floods occurred frequently in mid- and late July. In addition to tornadoes in Suihua, short-term heavy rain, strong winds, and hail battered the Aihui District of Heihe, Nenjiang County, Xunke County, Sunwu County, and Wulianchi. It can be seen from the disaster distribution map that the disasters in western China were more serious while the disasters in Heihe were relatively mild.

During the time phase DOY 225–DOY 241, disasters occurred in 2011, 2015, 2016, and 2017, and were relatively serious. In August 2010, heavy rains and floods developed frequently in Qiqihar and Hegang. In the provincial distribution map extracted on DOY 225, in addition to the above disasters consistent with the meteorological data, a large range of disasters were found in the eastern and northeastern sections of Heilongjiang Province. By the end of August 2011, a severe meteorological drought had developed in the eastern region, mainly in Mudanjiang, Harbin, Shuangyashan, Hulin, and other places, and especially in Linkou and Muling. These findings are consistent with the disaster range extracted in 2011 in Figure 5e. Meanwhile, it can be seen from Figure 5e that the Heihe River in the northwestern part of the province also experienced a serious disaster. At the end of August 2012, a windstorm caused large areas of crop lodging in cities and counties in the central part of Suihua and the Sanjiang Plain, resulting in serious urban waterlogging in Harbin. From the extracted disaster map, it can be seen that, with the exceptions of the disasters consistent with the above meteorological data, the flooding on the Sanjiang Plain was relatively serious. In the summer of 2013, Heilongjiang Province experienced heavy precipitation. In mid-August, Fuyuan County was stricken by floods and waterlogging and suffered serious losses, which was in agreement with the distribution map of extracted disasters across the entire province. Furthermore, the eastern part of Heilongjiang Province suffered from a large range of disasters. At the end of August 2016, strong winds and rainstorms hit the eastern part of the province. Gusts in Tongjiang even reached level 10; in Fuyuan, Suibin, Fujin, and Huachuan level 9; in Tonghe, Dongning, and 13 other counties and cities level 8; and in Suifenhe, Yilan, and 30 other counties and cities level 7. The high winds caused the partial lodging of rice and corn crops. The aforementioned observations were consistent with the extraction range. In August 2017, the amount of precipitation in Heilongjiang Province increased. In mid-August, a severe flood occurred in the city of Anda, and also took place along a number of small and medium-sized rivers, including the Tongkan, Hulan, Zhaolanxin, Belahong, Maolan, Dongxiao and Helen, with their water levels rising rapidly. The disaster distribution map of the entire province indicated that the flooding was serious on DOY 247.

On DOY 257, the disasters in 2010 and 2019 were still serious. In 2010, droughts occurred in Heilongjiang Province from late spring to early summer, and also in September. As seen in Figure 5f, the 2010 disaster map revealed that disasters mainly occurred on the Sanjiang Plain and in the eastern part of Heilongjiang Province. Since the fall of 2011, the continuous high temperatures and insufficient rainfall in Heilongjiang Province have led to meteorological drought in some areas. The disaster monitoring results extracted in 2011 primarily indicate drought in the east. In mid-September 2012, Typhoon "Sanba" tracked northward, disturbing the normal conditions in the eastern part of Heilongjiang Province. The associated precipitation from this system alleviated the previous drought and water shortage of reservoirs in the eastern part of Heilongjiang Province. Furthermore, the amount of precipitation in September was high. From the disaster scope extraction map, it can be seen that waterlogging resulted from serious river flooding. Since the rice crop was harvested early in some areas, however, the disaster range of the phase extraction was large [60].

It can be seen from Figures 5 and 6, Table A1, and the meteorological data analysis that, according to the distribution of disasters throughout the year, 2010, 2011, and 2012 were normal years, while the disasters in 2014, 2015, 2017, and 2018 were relatively mild, and those in 2013, 2016, and 2019 were serious.

By analyzing the meteorological disaster data, Figures 5 and 6, as well as the above discussion, we were able to summarize the spatial and temporal distribution characteristics of disasters from 2010 to 2019 in Heilongjiang Province. In terms of time distribution, disasters occurred frequently in July and August; spatially, disasters mainly took place in the central, eastern, and southwestern regions from June to August, including Qiqihar, Heihe, Suihua, Harbin, Jiamusi, and other locations.

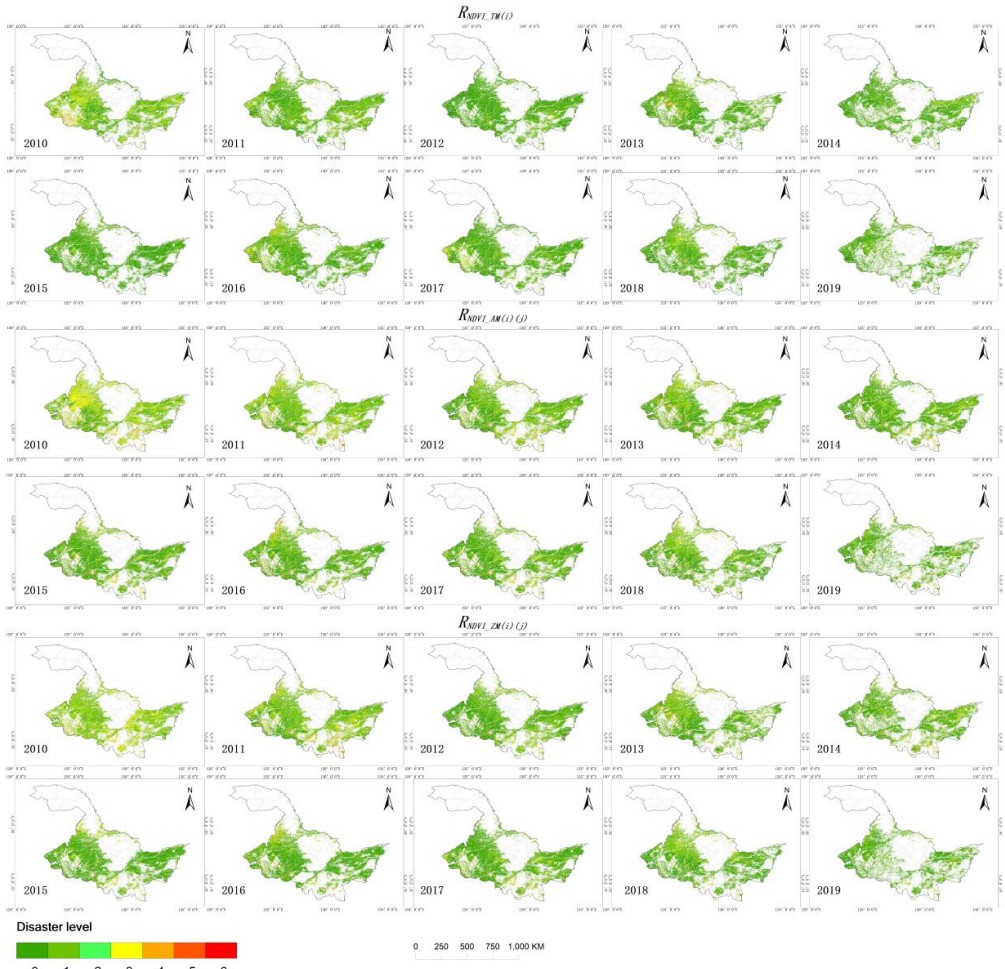

**Figure 6.** Disaster classification in Heilongjiang Province.

Different disasters exhibited different spatial and temporal distribution characteristics. Submersion was observed frequently in late June, although it also occurred in July and August, with the exception of the northwest Greater Khingan Mountains. In some years, submersion was prevalent in September. This type of disaster was primarily distributed in the northeast of Heilongjiang Province, in places such as Jiamusi, Tongjiang, Fuyuan, Fujin, and Suiling County in Hegang. In addition, Shuangyashan in the east, Qiqihar in the west, and Daqing and Suihua in the southwest were also frequently flooded. This is due to the fact that precipitation in Heilongjiang Province is concentrated from June to August, and the terrain is high in the northwest, north, and southeast, and low in the northeast and southwest. This means that, in terms of precipitation, a substantial difference exists between the eastern and western regions of Heilongjiang Province throughout the year, with large amounts of precipitation in the eastern and western regions and small amounts in the central and southern regions. The Songnen Plain and the Sanjiang Plain comprise higher topographical terrain and favorable water vapor conditions, making it easy for heavy precipitation to develop, and leading to numerous flood disasters. At the same time, since the central and northwest areas of the province are high while the northeast and west are low, flooding occurs readily.

Hail disasters occur frequently in June and July, and from late August to mid-September. Jiamusi in the east is a frequent disaster site, as are Shuangyashan, Mudanjian, and Jixi. In the western part of the province, hail disasters are concentrated in the Suihua, Heihe, and Qiqihar areas. The mountainous area represented by the Greater Hinggan Mountains experiences low temperatures and is prone to suffering from frost and hail disasters. These occur primarily as a result of orographic lifting and the planting structure of crops.

Droughts in Heilongjiang Province generally occurred in July and September, with the July droughts mainly developing in the southwest Suihua urban area and Harbin, as well as Daqing, in Durbert Mongolian Autonomous County, Zhaozhou County, and Zhaoyuan County. In the northwest, Baoqing County, Suibin County, Fujin, Tongjiang, and Fuyuan County are located on the Sanjiang Plain. In September, droughts mainly occurred in the northeast, including Yichun and Hegang in the north, as well as Lubei County, Suibin County, and Tongjiang. Qiqihar and the Mudanjiang area experienced high temperatures and were prone to drought.

In Heilongjiang Province, wind damage was always observed in August and September, while there were fewer windstorms in July. This type of disaster mainly occurred in the east and northeast areas, as well as in central and eastern regions such as Huachuan County, Suijiang County, Fujin, Tongjiang, Fuyuan County in Jiamusi, Yilan County, Shuangyashan in Jixian, Baoqing, Raohe, and Jidong County, as well as Jixi and Hulin.

## 4. Discussion

To date, many methods have been used to monitor agricultural disasters such as drought, waterlogging disasters, and so on. These methods have higher accuracy for single disasters [61–63]. At the same time, since research projects have tended to focus on small areas, there is still no rapid method for extracting disaster ranges from large areas. The emergence of GEE makes it possible to quickly extract large-scale agricultural disasters [45]. In general, the three disaster monitoring models exhibited high accuracy, although their monitoring accuracy levels for various disasters were different. The monitoring accuracy levels of hailstorms, droughts, and floods were higher. For insect and wind disasters, the real-time monitoring accuracy levels were low, and the phenomenon of disaster lag usually appeared in the subsequent images 16 days later. This is due to the fact that the disasters caused by hailstorms, floods, and droughts are immediate and serious for crops, with short duration and clear changes in the satellite images. The damage to crops from pests and windstorms, however, is continuous rather than short-term, and does not immediately cause changes in the images.

The accuracy levels of the disaster extraction range of different phases were also different. According to the extraction differences of the disaster ranges listed in Table 7, the disaster ranges on DOY 130, 145, 167, and 273 presented great differences, and the ranges themselves were large, with consistency only occurring from DOY 177 to DOY 257. This was mainly due to the low crop coverage and large bare soil area before mid-June; meanwhile, in September, when rice and other crops entered the tasseling stage, and some crops were premature, this led to the phenomenon of "no yield" on the image after the large area of rice was harvested. Therefore, the vegetation index of the three monitoring models in this area was relatively low. In late August, rice was harvested in advance in some areas of Heilongjiang Province, but the range was small, leading to a large disaster scope being extracted on DOY 241 in some small areas. On DOY 247, this range had expanded further, bringing an additional increase in the disaster extraction range error. Among the three models, the $R_{NDVI\_AM(i)(j)}$ was highly sensitive to bare soil, and the disaster ranges that could be easily extracted on DOA 177 and DOA 257 were relatively large.

Examining the applicability and consistency of the three monitoring models for different disasters, we discovered that the $R_{NDVI\_TM(i)}$ and $R_{NDVI\_ZM(i)(j)}$ displayed higher monitoring precision and a similar extraction range for hailstorms and windstorms; likewise for $R_{NDVI\_AM(i)(j)}$ and $R_{NDVI\_ZM(i)(j)}$ in terms of floods and droughts. Heilongjiang Province is vulnerable to flooding in August and September, resulting in a greater range of disasters than the $R_{NDVI\_TM(i)}$ extraction. This may be due to the varying mechanisms of the different monitoring models. At present, research on crop condition monitoring has primarily focused on multi-year comparisons based on the NDVI. The difference between the current value and the standard value is examined by taking the multi-year average value or the value of a specific reference year as the standard value for crop growth monitoring and disaster extraction [64]. This principle is thus the same as that of the $R_{NDVI\_TM(i)}$ model. This standard value is mainly reflected by the historical average crop growth. Its main disadvantage is that during a long

service life the crop planting structure may change, thus affecting the standard value. For example, in the research of Q. Huang et al., the NDVI value was compared with the average value of the NDVI for the previous five years, and the application and effect of the NDVI in spring wheat, winter wheat, spring corn, summer maize, cotton, soybean, and rice were investigated [38]; however, they failed to quantify the applicability and accuracy of different disasters in crop monitoring. In their research prospects, these scientists suggested that different remote sensing monitoring index systems should be established for different agricultural divisions. In fact, the $R_{NDVI\_AM(i)(j)}$ was proposed in terms of phenology and can effectively solve the above problems. By extracting the regional median value of different phenological regions as the standard value, the average growth situations of crops in various phenological areas are reflected, which are not affected by changes of crop planting structure. In other studies, the pNDVI has also been used to solve this problem [39], although the monitoring accuracy of different disasters has not been quantified. Compared with the $R_{NDVI\_AM(i)(j)}$, the $R_{NDVI\_ZM(i)(j)}$ cannot reflect the change of crop growth relative to the historical average. In order to solve this problem, we introduced the $R_{NDVI\_ZM(i)(j)}$, which not only reflects the comparison of crop growth level with the historical average level, but also reflects the average growth status of a given phenological region. The $R_{NDVI\_AM(i)(j)}$ and $R_{NDVI\_ZM(i)(j)}$ models were less affected by changes of planting structure. By comparing the applicability and accuracy of the three methods for different disasters, it was discovered that the accuracy was higher for hailstorms, droughts, and waterlogging. In addition, the model based on the GEE platform can be used for large-scale spatiotemporal pattern analysis and real-time monitoring. The results are basically consistent with the spatial and temporal distributions of drought and flood events based on the monthly precipitation data of meteorological stations using the standardized precipitation index (SPI), principal component analysis, Mann Kendall trend analysis, and Morlet wavelet analysis [57]. This method can be used to improve CropWatch because of the improvement of the uncertainty caused by the annual changes of crop rotation and phenology [64].

There are some common problems in the extraction of disaster scope by the three monitoring models, namely, their low spatial resolution results in the existence of mixed pixels, which in turn leads to the low detection accuracy of some small-scale agricultural disasters. Monitoring methods with higher spatial and temporal resolution can be adopted in order to improve the monitoring accuracy. Additionally, the growth period differences of different crop types were not fully considered in this study. In future research, the planting structure data for the entire province should be combined in order to perform further detailed analyses. In addition, investigations should continue to take advantage of the rapidity, wide range, and good portability of GEE, and expand the study area in an attempt to conduct disaster monitoring analyses of the global farmland scale or to compare the differences of disasters at the same latitude, thereby determining the underlying laws governing these events and the reasons for their occurrence. Higher-resolution images can be utilized, such as GF-1 high-resolution satellite imagery for agricultural remote sensing monitoring, as well as higher-resolution validation data [65]. This research provides technical support for disaster early warning, disaster prevention and mitigation, as well as post-disaster rescue work through the extraction of such large-scale and long-duration series of disaster scopes.

## 5. Conclusions

In this study, three models, i.e., $R_{NDVI\_TM(i)}$, $R_{NDVI\_AM(i)(j)}$, and $R_{NDVI\_ZM(i)(j)}$, were constructed using the GEE platform to extract the scope of disasters in Heilongjiang Province from 2010 to 2019. In addition, the spatiotemporal pattern changes and the applicability of the different models to various disasters were studied in combination with meteorological data. The results revealed the following:

1. The $R_{NDVI\_TM(i)}$, $R_{NDVI\_AM(i)(j)}$, and $R_{NDVI\_ZM(i)(j)}$ models could all extract the spatiotemporal features of large-scale disasters with high precision, which was consistent with the disaster situations and time variation trends reported across the entire province, and achieved the ideal result of disaster range extraction based on MODIS data.

2. The $R_{NDVI\_TM(i)}$, $R_{NDVI\_AM(i)(j)}$, and $R_{NDVI\_ZM(i)(j)}$ models were shown to have different applicability to hailstorms, floods, droughts, insect disasters, and windstorms, as well as different disaster extraction ranges. In addition, there was a strong consistency from DOY 177 to DOY 257, and the extraction disaster ranges were similar.

3. The disaster scopes extracted by the $R_{NDVI\_TM(i)}$, $R_{NDVI\_AM(i)(j)}$, and $R_{NDVI\_ZM(i)(j)}$ models were found to be in good agreement with the meteorological disaster data of Heilongjiang Province and can therefore be used to analyze the spatiotemporal pattern of disasters and to provide support for disaster risk partitioning.

**Author Contributions:** Conceptualization, Z.L. and H.L.; investigation, Z.L.; methodology, Z.L.; resources, C.L. and Y.J.; visualization, Z.L. and H.Y.; writing—original draft, Z.L.; writing—review & editing, X.M., H.L. and D.G. All authors have read and agreed to the published version of the manuscript.

**Funding:** This research was funded by the Special Foundation for Basic Research Program in Wild China of CAS, grant number XDA23070501.

**Conflicts of Interest:** The authors declare no conflict of interest.

**Appendix A**

**Table A1.** Extraction of disaster area ratio by three models in each time phase.

| Model. | Year | DOY 177 | DOY 193 | DOY 209 | DOY 225 | DOY 241 | DOY 257 |
|---|---|---|---|---|---|---|---|
| $R_{NDVI\_TM(i)}$ | 2010 | 0.25 | 0.03 | 0.04 | 0.07 | 0.39 | 0.35 |
| | 2011 | 0.22 | 0.03 | 0.02 | 0.03 | 0.11 | 0.23 |
| | 2012 | 0.16 | 0.02 | 0.01 | 0.02 | 0.04 | 0.04 |
| | 2013 | 0.03 | 0.13 | 0.06 | 0.06 | 0.10 | 0.20 |
| | 2014 | 0.03 | 0.03 | 0.03 | 0.05 | 0.18 | 0.11 |
| | 2015 | 0.12 | 0.05 | 0.02 | 0.02 | 0.02 | 0.02 |
| | 2016 | 0.05 | 0.06 | 0.03 | 0.02 | 0.08 | 0.10 |
| | 2017 | 0.11 | 0.06 | 0.03 | 0.03 | 0.08 | 0.15 |
| | 2018 | 0.08 | 0.02 | 0.02 | 0.02 | 0.08 | 0.23 |
| | 2019 | 0.07 | 0.03 | 0.02 | 0.08 | 0.10 | 0.09 |
| $R_{NDVI\_AM(i)(j)}$ | 2010 | 0.49 | 0.06 | 0.04 | 0.06 | 0.34 | 0.54 |
| | 2011 | 0.35 | 0.06 | 0.04 | 0.07 | 0.18 | 0.51 |
| | 2012 | 0.29 | 0.05 | 0.03 | 0.06 | 0.15 | 0.30 |
| | 2013 | 0.11 | 0.13 | 0.06 | 0.08 | 0.14 | 0.40 |
| | 2014 | 0.10 | 0.04 | 0.05 | 0.09 | 0.21 | 0.25 |
| | 2015 | 0.15 | 0.07 | 0.05 | 0.05 | 0.13 | 0.20 |
| | 2016 | 0.22 | 0.07 | 0.05 | 0.08 | 0.12 | 0.23 |
| | 2017 | 0.13 | 0.07 | 0.04 | 0.04 | 0.10 | 0.19 |
| | 2018 | 0.13 | 0.06 | 0.05 | 0.05 | 0.13 | 0.35 |
| | 2019 | 0.10 | 0.05 | 0.04 | 0.11 | 0.10 | 0.12 |
| $R_{NDVI\_ZM(i)(j)}$ | 2010 | 0.36 | 0.04 | 0.04 | 0.07 | 0.31 | 0.60 |
| | 2011 | 0.33 | 0.05 | 0.03 | 0.06 | 0.17 | 0.53 |
| | 2012 | 0.26 | 0.05 | 0.03 | 0.04 | 0.07 | 0.20 |
| | 2013 | 0.07 | 0.10 | 0.05 | 0.07 | 0.13 | 0.44 |
| | 2014 | 0.05 | 0.03 | 0.04 | 0.09 | 0.25 | 0.29 |
| | 2015 | 0.13 | 0.23 | 0.03 | 0.03 | 0.06 | 0.10 |
| | 2016 | 0.19 | 0.06 | 0.04 | 0.07 | 0.07 | 0.16 |
| | 2017 | 0.11 | 0.07 | 0.04 | 0.04 | 0.12 | 0.17 |
| | 2018 | 0.08 | 0.03 | 0.04 | 0.05 | 0.12 | 0.32 |
| | 2019 | 0.08 | 0.04 | 0.03 | 0.11 | 0.09 | 0.14 |

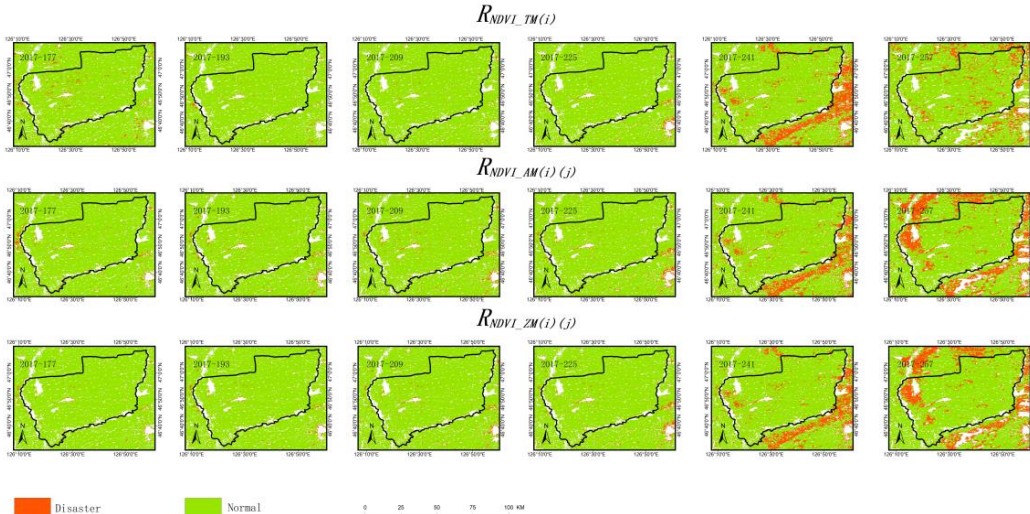

**Figure A1.** 2017 disaster distribution maps of Zhaokui County for the three monitoring models.

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
