# Peer review of "Rapid Extraction of Regional-scale Agricultural Disasters by the Standardized Monitoring Model Based on Google Earth Engine"

_sustainability, doi:10.3390/su12166497_

Round 1

Reviewer 1 Report

The thematic focus of the work has a current topic, but article still need to be modify and explain in more detailed way.

Recommendations for addition:

  1. Section 1. Introduction should include more detailed information on current needs and issues related to the use of GEE (or possible restrictions), for extraction of Agricultural Disasters, in order to highlight the novelty of the paper.
  2. The monitoring of agricultural disasters through vegetation NDVI data based on tools from the GIS environment also plays an essential role. I recommend the authors provide in Section 1. Introduction a very brief overview of the GIS application to the issue of monitoring agricultural disasters. In principle, GIS and Remote Sensing play an important role in the processing of issues. To avoid having to search for a long time, I offer an overview of a few posts on this issue: https://doi.org/10.3390/agriengineering2020017, https://doi.org/10.3390/min10060489
  3. Row 128 - If you use abbreviations for Google Earth Engine, then apply it for all article, Row 130...
  4. In some parts of article, there is no reference in the text to the presented images. For example subsection 2.1 – Figure 1, please apply it for all article.
  5. Subsection 2.2 Data is a very important part of processing, please extend individual subsections with more specific characteristics of the data types, e.g. in the form of tables. It means more specification for MODIS vegetation index (MOD13Q1), HJ-1A/B, and also Meteorological Data.
  6. Subsection 3.4.1. Spatial and temporal pattern analysis of 2017 disasters in Heilongjiang Province, it is not appropriate to state a picture immediately after the title of the subchapter, it is necessary to characterize the chapter in more detail, I recommend moving the paragraph from row 384 to the beginning of this subchapter, please apply it for all article.
  7. Figure 4 is illegible; please improve the quality of the presented image, or comment on the presented scenarios in more detail. Via the table (for individual presented DOY) is a possible way to present the range of individual disasters through the area of disaster.
  8. Figure 5 is illegible; I recommend change the form of presenting results.
  9. How did the authors provide a validation of their results?
  10. In section 4. Discussion, is needed a deeper discussion to evaluate and compare the achieved results with other scientific works of given importance.

Author Response

see atachment

Reviewer 2 Report

SUMMARY OF THE REVIEW:

The submitted paper aims to develop a rapid and effective method for monitoring agricultural disasters based on different NDVI standardization models by considering the median time, the median phenology, the median spatiotemporal standardization and combining MODIS data for large scale analyses. I found this paper very interesting where Several technical aspects were nicely implemented and explained. Undoubtedly, authors invested huge amount of time and have made a great effort to produce this high-quality of research which is clearly structured and the language used is largely appropriate. Nevertheless, I found few gaps that need correction and improvements. As final decision, I see that this manuscript in its form and level deserves to be accepted for publication in MDPI-S but after addressing below MINOR COMMENTS.

DETAILED COMMENTS PER SECTION:

  • The title is adequate for the content of the paper but can be improved to be more attractive…
  • For affiliations, please add the FULL AFFILIATION for each author including the post code of the region and the country.
  • ABSTRACT:
  • The abstract is well written and structured. Why Google Earth is not mentioned in the abstract?
  • List of keywords can be improved to cover the whole context of this nice paper.
  • INTRODUCTION:
  • The introduction is well structured and contain very important information about the background of the research and some technical aspects of the subject.
  • MATERIALS AND METHODS:
  • Map of the study area must be improved to match the high level & novelty of this paper.
  • I suggest to extend the section “2.3.6. Accuracy verification” and add more explanation.
  • RESULTS & DSICUSSION
  • I found these sections of the paper well written and include a very important finding.
  • Conclusion: Authors made a good conclusion and very interesting recommendations.
  • References: All the references MUST BE CHECKED and formatted as required by MDPI-S, also make sure that all the references have DOI number unless it is not available.

Author Response

See atachment

Reviewer 3 Report

Line 26: 2005 or 2010?

Second paragraph of introduction is too long?

Fig1: present with higher quality

How did you specify the cultivated land in Fig 1

Did you remove the cloud from the RS data? “Line 35: We selected the good data and marginal data from the SummaryQA in order to remove the impact of clouds and snow and ensure that the extracted disaster scope was not affected by outliers.” But which method?

Did you do preprocessing on the RS?

How many images did you used?

How did you evaluate the methods? Section: 2.3.6. Accuracy verification is not clear? You used MODIS data to evaluate the indices? Or you had ground truth data?

Please add text about Phenological zones in Heilongjiang Province in Figure 2. It is unclear?

Figure 4: better quality

Figure 5: better quality

Figure 6: better quality

I recommend after presenting the above figures in better quality, focus on small part of study area to zoom in, and show the variations better.

Round 2

Reviewer 3 Report

accept